

# Retrospective study on the association between paternal occupational exposure to agrochemicals and childhood leukemia in Michoacán de Ocampo, México

Paola Jiménez-Alcántar[1,*], Anel Gómez-García[2], Joel E. López-Meza[1], Alejandra Ochoa-Zarzosa[1], Luis Andrés Espino-Barajas[3], Luis Miguel Morales-Manilla[3], Eloy Pérez-Rivera[4], Luz Yadira Zúñiga-Quijano[4] and Sergio Gutiérrez-Castellanos[2,5,*]

[1] Centro Multidisciplinario de Estudios en Biotecnología-FMVZ, Universidad Michoacana de San Nicolás de Hidalgo, Morelia, Michoacán, México
[2] Centro de Investigación Biomédica de Michoacán, División de Investigación Clínica, Instituto Mexicano del Seguro Social, Morelia, Michoacán, México
[3] Centro de Investigaciones en Geografía Ambiental, Universidad Nacional Autónoma de México Campus Morelia, Morelia, Michoacán, México
[4] Hospital Infantil "Eva Sámano de López Mateos", Secretaría de Salud de Michoacán, Morelia, Michoacán, México
[5] Laboratorio de Citopatología Molecular, División de estudios de Posgrado, la Facultad de Medicina "Dr. Ignacio Chávez", Universidad Michoacana de San Nicolás de Hidalgo, Morelia, Michoacán, México
* These authors contributed equally to this work.

Corresponding author
Sergio Gutiérrez-Castellanos,
sergio.gutierrez@umich.mx

## ABSTRACT

**Objectives**. The constant use of agrochemicals in avocado plantations, because of their susceptibility to pests and diseases, continuously exposes those who work or reside near these orchards to health risks. The purpose of this study was to investigate the association between childhood leukemia cases in Michoacán, paternal occupational exposure to agrochemicals, and environmental exposure due to residential proximity.
**Methods**. A retrospective observational cohort study was performed. We analyzed 430 cases of leukemia in children under 18 years of age diagnosed between 2010-2023. Logistic regression models were used to estimate odds ratios and 95% confidence intervals, adjusted for sociodemographic factors. Survival was analyzed using Kaplan-Meier curves.
**Results**. A total of 46.6% of the parents of children with leukemia in this study had jobs related to the use of agrochemicals (*e.g.*, day laborers, peasants, farmers). Additionally, 65.4% of the leukemia cases occurred in municipalities producing avocado, the most important perennial crop in the state. Regarding the residential area analysis (mapping), many cases were found in contiguous zip codes and in areas densely occupied by avocado orchards. In addition, paternal occupations related to agrochemical use were associated with the avocado-growing zone, with an OR = 1.764 (95% CI [1.034–3.009], $p = 0.0379$). Interestingly, survival associated with agrochemical-related occupations has a higher mean survival (139.3 months) than all other parental occupations ($p = 0.0148$).
**Conclusions**. The epidemiologic evidence found in this study supports the association between paternal occupational exposure to agrochemicals and childhood leukemia.

Furthermore, children with leukemia who live in avocado-growing regions have a higher survival rate.

## INTRODUCTION

According to the National Institute of Public Health, childhood cancer is the leading cause of disease-related death among children aged 5 to 14 years in México, and the sixth leading cause among children under 5 years old. The most common types of childhood cancer include leukemia, brain tumors, lymphoma, and solid tumors such as neuroblastoma and Wilms' tumor. Acute lymphoblastic leukemia (ALL) is the most frequent malignant neoplasm among Mexican children and adolescents, with global survival estimates ranging from 40% to 60%. In contrast, in high-income countries, survival rates exceed 80% (*INSP, 2021*). Several large-scale epidemiological studies suggest that the incidence of ALL in México is among the highest in the world. While the global incidence ranges from 2 to 3.5 cases per 100,000 inhabitants, in México it exceeds six cases per 100,000. The reasons for this elevated incidence remain unknown (*Colunga-Pedraza et al., 2018*).

The etiology of leukemias has been explained by certain genetic conditions (*e.g.,* Down syndrome, Nijmegen syndrome) and some environmental conditions (*e.g.,* exposure to ionizing radiation, chemotherapeutics, and solvents). However, less than 5% of all cases can be explained by these causes (*Metayer et al., 2016*; *Valencia-González et al., 2021*). On the other hand, some observational epidemiological studies demonstrate a link between pesticide exposure and leukemia (*Roberts & Karr, 2012*). For example, prenatal maternal occupational exposure to pesticides, insecticides, and herbicides has been associated with the presence of childhood leukemia (*Karalexi et al., 2021*). Similarly, paternal exposure to pesticides has also been positively associated with an increased risk of childhood acute myeloid leukemia (AML) (*Patel et al., 2020*) as well as childhood acute lymphoblastic leukemia (ALL) (*Gunier et al., 2017*). Even residential proximity to areas where agrochemicals are used has been proposed as a risk factor for childhood ALL, associated with moderate pesticide exposure during life (*Rull et al., 2009*). However, there are also some studies in which no evidence of an association with maternal or paternal exposure to leukemia was found (*Van Maele-Fabry et al., 2010*; *Coste et al., 2020*), indicating that there is still a lack of conclusive evidence associating paternal occupational exposure to agrochemicals with the risk of childhood leukemia.

In the world ranking, México is the main avocado producer with 1.6 million tons, of which 994 thousand tons are exported (*Secretariat of Agriculture, Livestock, Rural Development, Fisheries, and Food (SAGARRA), 2024*; *Ministry of Agriculture and Rural Development (SADER), 2023*). In the country, Michoacán is the leading avocado producer, producing 74% of all Mexican avocado (*Cruz-López et al., 2022*). Avocado crop variants are susceptible to pests and diseases that affect fruit growth, so many agrochemicals are

used, including pesticides such as insecticides, herbicides, fungicides, and nematicides for pest and weed control (*Borrego & Carlón-Allende, 2021*). In México, there are 183 authorized pesticide active ingredients, 34% of which have been reported to exhibit high acute toxicity, and 23% of which are associated with chronic toxicity concerns (*Bejarano-González et al., 2017*). In the specific case of avocado cultivation, the Association of Producers and Exporting Packers of México A.C. (APEAM) defines the active pesticide substances permitted for avocado crops regulated for export (*Rodríguez-Campos, Escobedo-Reyes & Lugo-Cervantes, 2017*). This list includes 18 fungicides, notably folpet, which has been classified as a possible carcinogen by the Environmental Protection Agency; however, the product information has not yet been updated. The list also includes 25 insecticides, among which malathion is widely used and has similarly been classified as a possible carcinogen. Studies have even shown that it induces cancer-related gene expression in human lymphocytes (*Anjitha et al., 2020*). *In vitro* studies on malathion and permethrin—other notable insecticides on this list—indicate that both induce aberrations in genes involved in the etiology of hematological cancers, such as leukemia. Additionally, permethrin has been shown to induce aneuploidy, a common chromosomal abnormality in certain types of leukemia (*Navarrete-Meneses et al., 2017*). Furthermore, both insecticides have been found to cause epigenetic modifications in hematopoietic tissues, leading to the deregulation of genes potentially associated with the onset and progression of hematological cancers (*Navarrete-Meneses et al., 2023*). Regarding the herbicides on the APEAM list, five active substances have been identified. Among these, glyphosate stands out as the most widely used due to its broad-spectrum action. It has also been classified as a probable carcinogen by the International Agency for Research on Cancer (*IARC, 2015*), although its impact remains controversial. In Michoacán, a study conducted in the eastern avocado-growing zone of the state revealed that farmers use various types of pesticides, including insecticides (55%), herbicides (11%), and fungicides (33%), resulting in the use of 28 active ingredients. This list includes benomyl and paraquat, which are banned internationally but still permitted in México. Additionally, imidacloprid and glyphosate are among the most used substances. Of all the pesticides reported, 80% are classified as highly hazardous according to the 2021 International List of Highly Hazardous Pesticides (HHPs) published by the Pesticide Action Network (PAN) (*Merlo-Reyes et al., 2024*). The region where this research was conducted represents only a small fraction of the avocado-producing area in the state, and no further studies on this topic have been identified.

Currently, there are no reports in the state of Michoacán on the impact of agrochemicals in avocado orchards. We propose that the use of agrochemicals in avocado orchards could be related to leukemia cases in the state of Michoacán. Therefore, this study aimed to identify the association between childhood leukemia cases, paternal occupation related to agrochemicals, and residential proximity to avocado orchards.

## MATERIALS AND METHODS

### Study design

A retrospective observational cohort study was conducted. Medical records and death certificates from the period 2010 to 2023 were reviewed. All medical records of patients
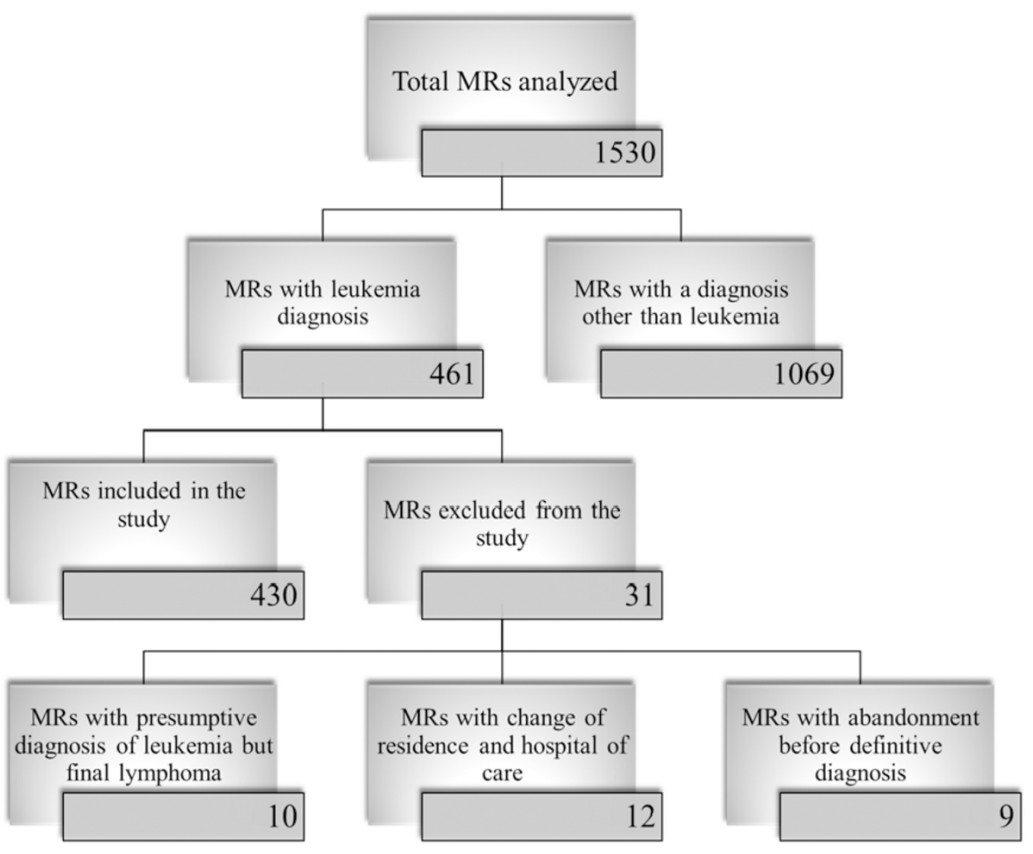

**Figure 1** **Selection of Participating Medical Records.** The flowchart shows the description of the selection process of the medical records from the 'Eva Sámano de López Mateos' Children's Hospital. MR: medical records.

with leukemia under 18 years of age from the oncology department of the Children's Hospital 'Eva Sámano de López Mateos' located in Michoacán de Ocampo, México (Fig. 1), were included. The medical records were analyzed once without subsequent follow-up; this review was conducted from January to April 2024. In Michoacán, oncology care—including the treatment of childhood leukemia—is concentrated in three healthcare centers: the Mexican Social Security Institute (IMSS) and the Institute of Security and Social Services for State Workers (ISSSTE), which together manage between 10% and 15% of childhood leukemia cases in the state; and the Children's Hospital "Eva Sámano de López Mateos", which accounts for 85% to 90% of all leukemia cases in Michoacán. It is important to note that there are no specialized hospitals in the various municipalities of the state.

Additionally, death certificates of children (under 18 years of age) whose underlying cause of death (ICD-10) was leukemia and whose care was provided at the Children's Hospital 'Eva Sámano de López Mateos' were included to complete the follow-up of medical records without outcome. These records were obtained from the 'Health Intelligence Center' (CIS) of the Ministry of Health of Michoacán de Ocampo, México. The CIS collects and manages

all mortality data for monitoring, health planning, and evaluating public health policies. These records were reviewed from March to May 2023.

The following data were collected from the medical records and death certificates for each patient and their parents: patient sex, age, occupation (paternal and maternal), education level (paternal and maternal), residential area, type of leukemia, risk, medical course, survival months and/or date of death with the causes of death. Overall survival was defined as the time from diagnosis to death or until the last progress note in the medical record for those patients not listed in the death registry. The causes of death obtained were recorded according to the International Statistical Classification of Diseases and Related Health Problems (ICD-10) by the WHO. Age groups were classified as high risk (under 1 year and from 10 to 17 years) and standard risk (from 1 to 9 years).

This research project was carried out in accordance with the ethical standards of the General Health Law Regulations on Health Research (*Cámara de diputados, 2014*). According to Article 17 of this regulation, this research is classified as type I. Risk-free research: retrospective documentary studies without intervention in physiological, psychological, and social variables, such as questionnaires, interviews, and review of medical records, without identification of subjects or treatment of sensitive behavioral aspects; therefore, informed consent is not required. A letter of exemption from informed consent, detailing data collection and confidential data management, was submitted to the Ethics Committee of the Children's Hospital of Morelia "Eva Sámano de López Mateos" (CI/10/2023) along with the research protocol for approval. The protocol was also submitted to the Local Health Research Committee 1602 of the Hospital General Regional No. 1 with register R-2023-1602-060.

This protocol was developed following the guidelines for strengthening the reporting of observational studies in epidemiology (STROBE).

## Standardized incidence ratio (SIR)

To assess the incidence of leukemia, the state was stratified into two zones: avocado-growing and non–avocado-growing. The cumulative incidence per 100,000 children (0–17 years) over a 10-year period was calculated for each zone using population census data from 2010 and 2020—the two available censuses within the study period—provided by the National Institute of Statistics and Geography (INEGI). Calculations were based on the number of observed cases and the average child population.

Calculation of expected cases and standardized incidence ratio (SIR). The expected number of cases for each zone and municipality was estimated by multiplying the corresponding average child population (0–17 years) by the state leukemia incidence rate. The state incidence rate was obtained by dividing the total number of cases recorded in the state by the total state population, expressed per 100,000 person-years, with population data obtained from INEGI. Subsequently, the SIR was calculated as the ratio between the observed number of cases and the expected number of cases in each zone or municipality. An SIR value of 1 indicates that the observed incidence matches the expected incidence, values greater than 1 indicate that more cases were observed than expected, and values less than 1 indicate fewer observed cases than expected. To assess the precision of the SIR,

95% confidence intervals (CIs) were calculated under the assumption that events follow a Poisson distribution.

## Maps

The delimitation for the avocado agricultural frontier was updated from the one obtained by *Morales-Manilla et al. (2011)*, using information from Climate Rights International (*CRI, 2023*) certified orchards for export. Additionally, a 2015 land use map (*Benítez-Franco, 2021*) and a map of avocado orchards in eastern Michoacán (2020) (*López-Sánchez, 2024*) were used. Very high resolution (ESRI, Google Earth Pro) and high resolution (Sentinel-2, February–March 2024) images were utilized. The maps were created with ArcGIS Pro. Zip codes were downloaded in Shapefile format and a join was performed, defining ranges of leukemia cases and assigning color palettes according to the number of cases.

Based on the delineation of the state's avocado agricultural frontier, the analyses in this study were conducted by dividing the area into two zones: an avocado-growing zone and a non–avocado-growing zone. Additionally, the maps reflect the division of the state by municipalities and zip codes. In México, a "municipality" refers to the administrative subdivisions within each state, functioning as local government units.

## Statistical analysis

A Kolmogorov–Smirnov test was performed to estimate the normality of the distribution of the parameters investigated. The age variable had an abnormal distribution, data were expressed as median (minimum-maximum value). Categorical data were reported as percentages (%), these variables include age, sex, leukemia by cell lineage, paternal and maternal age, paternal and maternal education level, paternal and maternal occupation, survival (primary outcome), deaths, dropouts. Since the study is based on the review of medical records, there are some missing data. For this reason, the sample size (n) of the data used is specified in each table and in the description of the results. A Chi-square test was used to analyze differences between categorical variables. A logistic regression analysis was performed to investigate the relationship between paternal occupation, municipality of residence, and leukemia; the analysis was adjusted for age and sex. Municipalities of residence can act as a confounding variable; therefore, it was included in the analysis to minimize its impact as a source of confusion. Survival was analyzed using Kaplan–Meier curves. The significance of the difference observed between the various curves was determined by the Log Rank test. Additionally, a Cox proportional hazards regression was performed to assess the relative risk among four groups defined by the combination of municipality of residence and paternal occupation. Statistically significant differences were established when $P$ value $< 0.05$. The data were analyzed in the SPSS version 23 statistical package and the graphs were made in the GraphPad Prism 10 program.

## RESULTS

### Demographic information

Data were collected from a total of 430 medical records of leukemia and death records from 2010 to 2023 at the "Hospital Infantil Eva Sámano de López Mateos" in Morelia and the

**Table 1** Demographic and clinical characteristics of the pediatric population with leukemia under study.

| Characteristics | n | % |
| --- | --- | --- |
| Age | | |
|     Under 1 year old | 7 | 1.8 |
|     1 to 9 years old | 256 | 67.8 |
|     10 to 17 years old | 130 | 30.4 |
| Sex | | |
|     Female | 199 | 46.3 |
|     Male | 231 | 53.7 |
| Leukemia by cell lineage | | |
|     Myeloid | 50 | 11.7 |
|     Lymphoid | 380 | 88.3 |
| Immunophenotype of Lymphoid Leukemias | | |
|     B Cells | 361 | 95.6 |
|     T Cells | 17 | 4.4 |
| Immunophenotype of Myeloid Leukemias | | |
|     AML | 37 | 71.2 |
|     AML M3 | 9 | 17.3 |
|     CML | 6 | 11.5 |

Notes.
AML, Acute Myeloid Leukemia, CML, Chronic Myeloid Leukemia, M3, Fab classification for Acute Promyelocytic Leukemia.

''Centro de Inteligencia en Salud'', both located in the city of Morelia, Michoacán, México. The description of the inclusion and exclusion criteria for the medical records used in this study is detailed in Fig. 1. In this study, the medical records were analyzed once without subsequent follow-up. The complete records (up to the last recorded follow-up note) were reviewed, and for those records without an outcome, all children's death records from the Health Intelligence Center were reviewed. These records had leukemia as the underlying cause of death (ICD-10) and were treated at the 'Eva Sámano de López Mateos' Children's Hospital.

The demographic and clinical characteristics of the pediatric population with leukemia are shown in Table 1. 67.8% of the population presented the disease in the age range of 1 to 9 years (based on 393 located medical records), with a median age of 6 years. The classification of the age groups was based on the risk assignment for the disease by age (high risk, under 1 year and from 10 to 17 years and standard risk, from 1 to 9 years). Regarding sex (located medical records 430), 46.3% of the population was female and 53.7% was male. Most cases (88.3%) were of lymphoid lineage, with the predominant immunophenotype being B cells at 95.6%. In cases of myeloid leukemia, acute myeloid leukemia (AML) predominated, accounting for 71.2% of cases (located medical records of leukemia categorized by cell lineage, 430).

An analysis was conducted to determine whether there was a relationship between sex and leukemia type by lineage. Among males, 203 cases of lymphoid leukemia (88%) and 29 cases of myeloid leukemia (12%) were identified. Among females, there were 174 cases of lymphoid leukemia (89%) and 22 cases of myeloid leukemia (11%). A Pearson

**Table 2  Distribution of childhood leukemia cases according to paternal occupation related to agrochemicals and residential zone.**

| Characteristics | | Avocado-growing zone | | | | Non-avocado-growing zone | | | |
|---|---|---|---|---|---|---|---|---|---|
| | | Occupation related to agrochemicals | Other occupation | $\chi^2$ | $p$ | Occupation related to agrochemicals | Other occupation | $\chi^2$ | $p$ |
| Age | Less than 1 year old | 1 | 0 | 1.37 | 0.242 | 0 | 2 | 2.841 | 0.092 |
| | 1 to 9 years | 60 | 83 | – | Ref. | 40 | 27 | – | Ref. |
| | Over 10 years old | 17 | 31 | 0.639 | 0.424 | 22 | 15 | 0.001 | 1.0 |
| Sex | Male | 42 | 61 | 0.002 | 0.963 | 32 | 20 | 0.391 | 0.532 |
| | Female | 36 | 53 | | | 30 | 24 | | |
| Leukemia by cell lineage | Myeloid | 70 | 100 | 0.073 | 0.786 | 56 | 35 | 2.461 | 0.117 |
| | Lymphoid | 8 | 13 | | | 6 | 9 | | |

**Notes.**
$\chi^2$ = chi-square test; $p = \leq 0.05$. "Ref." indicates the reference category for comparisons.

chi-square test was performed ($\chi^2 = 0.300$, $p = 0.861$), indicating no statistically significant association between sex and leukemia type.

An analysis was also conducted to examine the relationship between age, sex, and leukemia type with paternal occupation (related or unrelated to agrochemicals) in avocado-growing and non-avocado-growing areas (Table 2). In both regions, most cases were concentrated in the 1–9 years age group, followed by individuals over 10 years of age and, lastly, infants under 1 year old. No statistically significant differences were found between age groups ($\chi^2 = 0.639$, $p = 0.424$ in the avocado-growing area; $\chi^2 = 0.001$, $p = 1.0$ in the non-avocado-growing area). Regarding sex, in the avocado-growing area, 42 cases were recorded in males and 36 in females whose fathers had occupations related to agrochemicals. In contrast, among fathers with other occupations, 61 cases were found in males and 53 in females. No statistically significant differences were observed between groups related or unrelated to agrochemicals ($\chi^2 = 0.002$, $p = 0.963$). Lastly, when comparing leukemia by cell lineage, in the avocado-growing area, 70 cases of lymphoid leukemia were recorded among children of exposed workers to agrochemicals and 100 among children of non-exposed workers. For myeloid leukemia, eight and 13 cases were recorded, respectively, with no statistically significant difference ($\chi^2 = 0.073$, $p = 0.786$). In the non-avocado-growing area, 56 cases of lymphoid leukemia were identified among exposed individuals and 35 among non-exposed individuals, while for myeloid leukemia, 6 and 9 cases were reported, respectively, with no significant association ($\chi^2 = 2.461$, $p = 0.117$). These results indicate that there is no statistically significant relationship between paternal occupation and age, sex, or leukemia lineage in any of the evaluated areas.

In terms of the demographic characteristics of the parents related to maternal occupation (available in 353 medical records) (Table 3), household-related activities predominated, accounting for 79.6%. Only 3.4% of mothers were engaged in occupations involving agrochemicals, and 1.4% in work related to paints and solvents.

As for paternal occupation (available in 333 medical records), 46.6% reported employment related to agrochemicals, including farming, day labor, and subsistence

**Table 3 Demographic characteristics of the parents.**

| Characteristics | Maternal | | Paternal | |
|---|---|---|---|---|
| | *n* | % | *n* | % |
| Median age (CI 95%) | 32 (31.08–32.92) | | 34 (32.49–35.51) | |
| Education level (%) | | | | |
| Illiterate | 20 | 6.3 | 18 | 6.4 |
| Primary | 120 | 37.9 | 135 | 48.0 |
| Secondary | 119 | 37.3 | 78 | 27.8 |
| Preparatory | 46 | 14.0 | 36 | 12.8 |
| University | 14 | 4.4 | 14 | 5.0 |
| Occupational activity (%) | | | | |
| Home related activities | 281 | 79.6 | 0 | 0 |
| Agrochemicals related activities | 12 | 3.4 | 155 | 46.6 |
| Construction related activities | 0 | 0 | 42 | 12.5 |
| Paints and solvents related activities | 5 | 1.4 | 22 | 6.5 |
| Other activities | 5 | 1.4 | 114 | 34.4 |

agriculture. However, information on the type of agrochemical exposure, as well as the intensity and duration, was not available in the medical records. Additionally, 12.5% of fathers worked in construction, and 6.5% were involved in activities related to paints and solvents (Table 3). Of the total number of cases where both paternal and maternal occupations were recorded (353), only 2.2% (8 cases) involved both parents working in activities related to agrochemicals. Occupations classified as "Other", along with details of the previously mentioned activities, can be found in Table S1.

## Frequency, incidence and risk of leukemia in the avocado-growing area

In Michoacán, avocado is the most important perennial crop in terms of cultivated area (*INEGI, 2021*), and based on this, we assessed whether the residence (near or in the same place as the orchards) of the population with leukemia under study was related to the frequency of the disease. For this purpose, the state was grouped into avocado-growing zone of residence and non-avocado-growing zone of residence. The avocado-growing zone included those municipalities in the state that are registered in the "Servicio de Información Agroalimentaria y Pesquera" (SIAP, report 2023) as avocado producers (see Table S2). Based on this information, 65.4% of cases (246) were found in children residing in the avocado-growing area, 28.5% (107 cases) in the non-avocado-growing area, and 6.1% (23 cases) in children with leukemia residing outside the state. To assess potential differences in case incidence between the studied areas, a SIR, or Observed/Expected (O/E) ratio. In the avocado-growing area, with a cumulative average pediatric population of 858,773, the 10-year incidence rate was 28.6 cases per 100,000 children, with 227.3 expected cases and an SIR of 1.08 (95% CI [0.951–1.226]). In the non-avocado-growing area, with 470,891 children, the incidence rate was 22.7 cases per 100,000 children, with 124.6 expected cases and an SIR of 0.85 (95% CI [0.703–1.03]).

**Table 4 Childhood leukemia incidence and standardized ratios in avocado-growing and non-avocado areas.**

| Zone | Observed cases | Average child population | 10-year incidence (x100,000) | Expected cases | Ratio O/E (SIR) | IC 95% |
|---|---|---|---|---|---|---|
| Avocado-growing zone | 246 | 858,773 | 28.6 | 227.3 | 1.08 | 0.951–1.226 |
| Non-avocado area | 107 | 470,891 | 22.7 | 124.6 | 0.85 | 0.703–1.03 |
| Total | 353 | 1,329,664 | 51.3 | 351.9 | 1.00 | 0.901–1.113 |

Notes.
O/E, Observed/Expected Ratio; SIR, Standardized Incidence Ratio.

**Table 5 Logistic regression analysis of the association between parental occupation and residence in the avocado-growing area.**

| | Occupational activity | OR | CI 95% | p |
|---|---|---|---|---|
| Paternal occupation n = 298 | Other activities | 1 | – | Ref. |
| | Agrochemicals related activities | 1.764 | 1.034–3.009 | 0.037 |
| | Construction related activities | 0.536 | 0.212–1.350 | 0.186 |
| | Paints and solvents related activities | 0.792 | 0.263–2.388 | 0.679 |
| Maternal occupation n = 287 | Other activities | 1 | – | Ref. |
| | Agrochemicals related activities | 1.515 | 0.769–2.988 | 0.230 |
| | Home related activities | 0.589 | 0.111–3.133 | 0.535 |
| | Paints and solvents related activities | 2.357 | 0.301–18.443 | 0.414 |

Notes.
p = ≤ 0.05. OR, odds ratio; "Ref." indicates the reference category for comparisons. OR were estimated using models adjusted for age and sex.

When both regions were combined, the resulting SIR was 1.00 (95% CI [0.901–1.113]) (Table 4). The 95% confidence interval analysis indicates that none of the evaluated areas showed statistically significant differences compared to the expected number of cases under a Poisson distribution.

In addition, the relationship between parental occupation and residence in the avocado-growing area was evaluated using logistic regression analysis, adjusting for age and sex. Paternal occupations related to agrochemical use were associated with the avocado-growing zone, with an OR = 1.764 (95% CI [1.034–3.009], p = 0.0379). Maternal occupations related to agrochemical use were not associated with the avocado-growing area, with an OR = 1.515 (95% CI [0.769–2.988], p = 0.230). The reference category consisted of other types of occupations (OR = 1). The other paternal or maternal occupations are reported in Table 5. None of the maternal occupation categories yielded statistically significant results (Table 5), suggesting that maternal occupation does not significantly influence the likelihood of residing in an avocado-growing area.

## Survival

Of the total cases (Table 3), 68% are alive, 24% died, and 8% abandoned treatment. Medical records of live patients indicate that 70% are under surveillance, 25% are in treatment, 4% had a relapse and are still in treatment, and 1% were discharged from the oncology service. Table 6 shows the breakdown of live, relapsed, and deceased cases, as well as the evolutionary status by type of leukemia. Age groups classified as high risk (under 1 year

**Table 6 Status of leukemia patients in the study according to the latest progress note in the medical record.**

| Status | B-ALL | | T-ALL | | AML | | CML | |
|---|---|---|---|---|---|---|---|---|
| | n | % | n | % | n | % | n | % |
| Deceased | 68 | 22 | 3 | 18 | 20 | 44 | 0 | 0 |
| Abandonment | 26 | 8 | 1 | 6 | 3 | 7 | 1 | 17 |
| Alive | 216 | 70 | 13 | 76 | 22 | 49 | 5 | 83 |
|     Treatment | 55 | 25 | 5 | 38 | 1 | 5 | 3 | 60 |
|     On treatment after relapse | 8 | 4 | 0 | 0 | 3 | 13 | 0 | 0 |
|     Surveillance | 151 | 70 | 8 | 62 | 17 | 77 | 2 | 40 |
|     Discharge for cure | 2 | 1 | 0 | 0 | 1 | 5 | 0 | 0 |

**Notes.**

ALL, Acute lymphoblastic leukemia of B-cell (B ALL) or T-cell (T ALL); AML, acute myeloid leukemia; CML, chronic myeloid leukemia.

and over 10 years) together accounted for 56.1% of deaths, while the age group from 1 to 9 years (standard risk) accounted for 43.9% of deaths. Additionally, the observed frequency of mortality during remission induction chemotherapy was 5.3% ($n = 25$). The first cause of death recorded in the CIS was septic shock or sepsis (45%), followed by hemorrhage (respiratory tract or intracranial) (16%), and leukemia as the base cause was the third with 8% of the cases (Table S3).

A statistical analysis was conducted to evaluate the relationship between survival status (alive or deceased) and sex among the studied cases. A total of 365 records were analyzed, showing that among males, 134 were alive (68.4%) and 62 were deceased (31.6%), while among females, 108 were alive (64.3%) and 60 were deceased (35.7%). Pearson's chi-square test indicated no statistically significant association between sex and survival status ($\chi^2 = 1.181$, $p = 0.554$).

Regarding survival by zone of residence, there is no statistically significant difference between survival in patients from the avocado-growing zone and patients from the non-avocado-growing zone. The estimated survival by zone was 121.8 months for the avocado-growing zone, 120.4 months for the non-avocado-growing zone, and 132.4 months for those living in other states of the Republic (Fig. 2A). In contrast, survival as a function of occupation shows a mean survival of 139.3 months for patients whose parents are engaged in agrochemical-related activities, 107.043 months for those whose parents are involved in construction-related activities, 89.2 months for children whose parents have an occupational activity related to paints and solvents, and 117.078 months for those whose parents have another occupational activity (Fig. 2B). The log-rank test indicates a significant difference between the survival curves ($p = 0.0148$), suggesting that survival appears to be more favorable for leukemia patients whose parents are engaged in some activity related to agrochemicals.

Additionally, a Cox regression analysis was conducted to evaluate the relative risk associated with combinations of residence zone and parental occupation. Occupation was grouped into activities related to agrochemicals and other types of activities. The overall model was not statistically significant, presenting a -2LL (log-likelihood) value

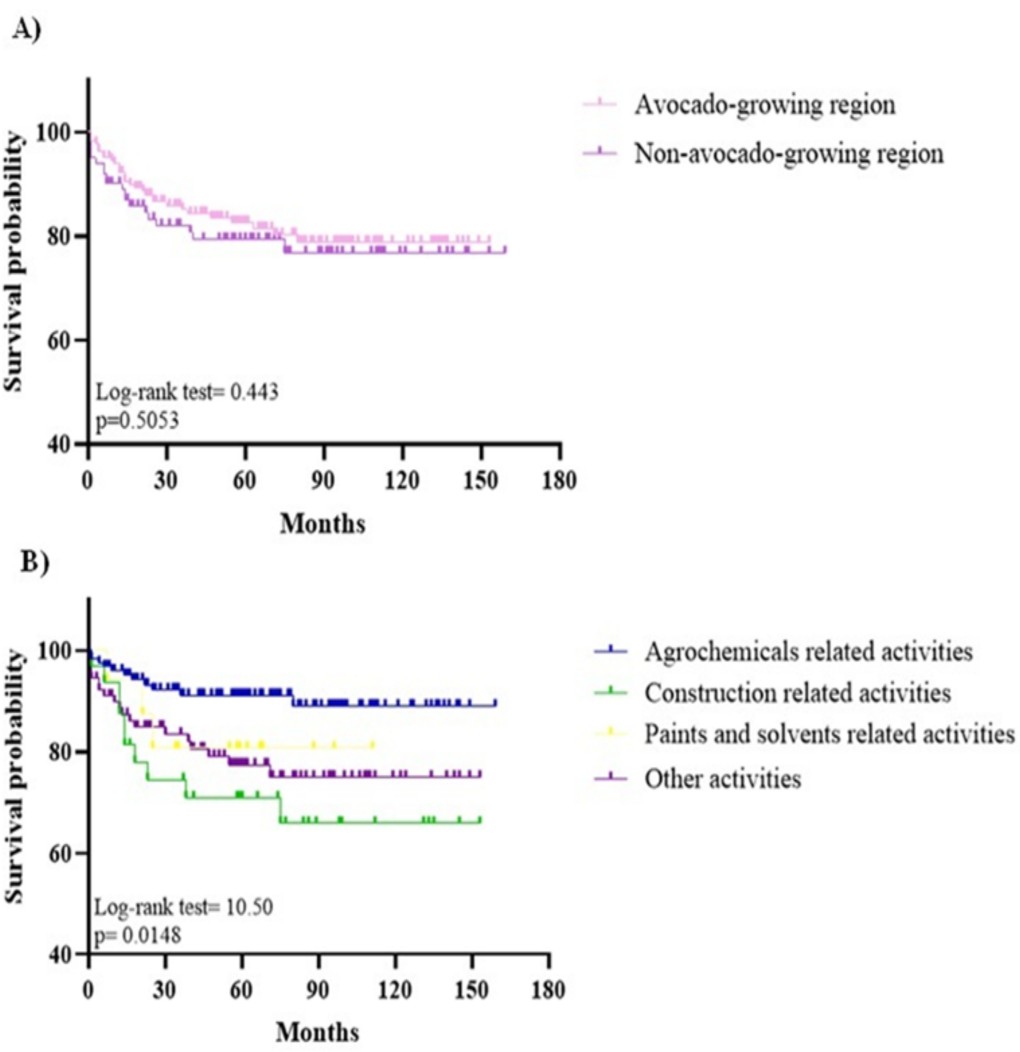

**Figure 2 Survival.** (A) Survival as a function of area of residence. The estimated survival by zone was 121.8 months for the avocado-growing zone, 120.4 months for the non-avocado-growing zone. (B) Survival as a function of paternal occupation. Survival as a function of occupation shows a mean survival of 139.3 months for patients whose parents are engaged in agrochemical-related activities, 107.043 months for those whose parents are engaged in construction-related activities, 89.2 months for children whose parents have an occupational activity related to paints and solvents, and 117.078 months for those whose parents have another occupational activity ($p = 0.0148$).

of 817.502, $p = 0.130$. The study population was divided into four groups, yielding the following results: children residing in the avocado-growing zone with parents engaged in agrochemical-related activities (HR = 0.508, 95% CI [0.248–1.040], $p = 0.064$); avocado zone resident children with parents engaged in other activities (HR = 0.840, 95% CI [0.459–1.538], $p = 0.572$); non-avocado zone resident children with parents engaged in agrochemical-related activities (HR = 0.521, 95% CI [0.244–1.112], $p = 1.112$); while children residing in non-avocado-growing zones with parents engaged in other activities

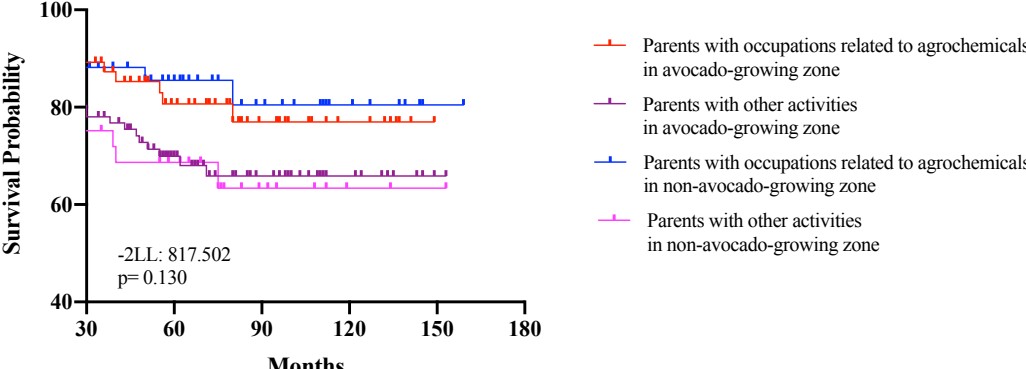

**Figure 3** **Hazard ratios for survival in children with leukemia according to paternal occupation and area of residence.** Survival curves stratified by parental occupation and avocado-growing zone. Children residing in the avocado-growing zone with parents engaged in agrochemical-related activities (red), avocado zone resident children with parents engaged in other activities (purple), non-avocado zone resident children with parents engaged in agrochemical-related activities (blue), and children residing in non-avocado-growing zones with parents engaged in other activities (pink) (reference group). Statistical adjustment values are included: (-2LL = 817.502), ($p = 0.130$).

(reference group). These results suggest that, overall, the combinations of residence zone and parental occupation are not significantly associated with survival (Fig. 3, Table 7).

## Mapping by municipality of residence and ZIP codes

On the other hand, based on the municipality of residence of the leukemia cases studied (from medical records 352), a map of Michoacán de Ocampo, México was created (Fig. 4 and Table 8), showing the 113 municipalities and leukemia cases using a color code. The avocado agricultural frontier (black line) and avocado growing areas (dotted in green) were delimited. The avocado-growing fringe comprises 65 municipalities with a frequency of 60% of all cases, consistent with the 2023 SIAP register (65.4%) (SIAP, 2023). The most representative municipalities are Morelia (45 cases), Uruapan (19), Hidalgo (14), Huetamo (13), Tacámbaro (13), and Apatzingán (10). By grouping, in the avocado-growing zone there are 2 municipalities with more than 15 cases, 2 municipalities with 9 to 14 cases, 2 municipalities with 6 to 8 cases, 55 municipalities with 1 to 5 cases, and 4 municipalities with no cases. In the zone outside the avocado frontier, there are 19 municipalities with no cases, 2 municipalities with 6 to 8 cases, 3 municipalities with 9 to 14 cases, and 24 municipalities with 1 to 5 cases. The Standardized Incidence Ratio (SIR) or Observed/Expected (O/E) ratio, based on a Poisson distribution, was also calculated for each municipality in the state. However, since most municipalities reported fewer than 10 cases, the SIR estimates in these areas are not statistically reliable, which limits their interpretation and validity (Table S4). Nevertheless, some municipalities showed a higher number of observed cases than expected. Specifically, the municipalities of Coeneo, Cherán, Jiménez, Tacámbaro, Tzintzunzán, and Zinapécuaro in the avocado-growing zone, and Huetamo and Múgica in the non-avocado-growing zone, all presented SIR values greater than 2. These findings may

**Table 7** **Hazard ratios for survival in children with leukemia according to paternal occupation and area of residence.**

| Area of residence | Paternal occupation | HR | 95% CI | *p* |
|---|---|---|---|---|
| Avocado-growing zone | Parents with occupations related to agrochemicals | 0.508 | 0.248–1.040 | 0.064 |
| | Parents with other activities | 0.840 | 0.459–1.538 | 0.572 |
| Non-avocado-growing zone | Parents with occupations related to agrochemicals | 0.521 | 0.244–1.112 | 1.112 |
| | Parents with other activities | 1 | – | Ref. |

**Notes.**

HR, Hazard ratio; CI 95%, Confidence Interval; $p \leq 0.05$. HR were estimated using models adjusted for age and sex.

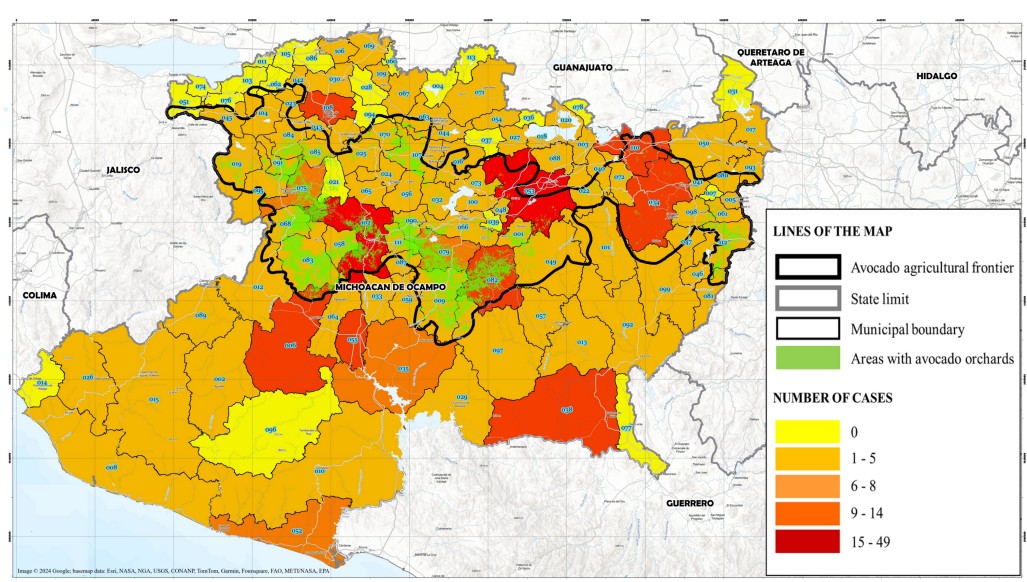

**Figure 4** **Distribution of childhood leukemia cases by zip code in Michoacán.** Sociodemographic data were obtained from medical records of pediatric leukemia patients treated at the "Eva Sámano de López Mateos" Children's Hospital between 2010 and 2023. Metadata prepared by the University Drone Laboratory (LUD), Center for Research in Environmental Geography (CIGA). Map adapted by the author. All rights reserved by the respective data providers. Map generated using Google Earth Pro (Image ⓒ2024 Google; basemap data: Esri, NASA, NGA, USGS, CONANP, TomTom, Garmin, Foursquare, FAO, METI/NASA, EPA).

suggest areas of potential epidemiological interest, although they should be interpreted with caution due to small sample sizes and wide confidence intervals (Table S4).

Finally, Fig. 5 shows the areas of residence of leukemia cases by zip codes (located medical records 307). Within the avocado agricultural frontier, cases are clustered in areas with a high density of avocado orchards. In Uruapan (municipality 102, Fig. 4), 13 of the 19 cases are concentrated in contiguous zip codes; between Pátzcuaro (066) and Salvador Escalante (079), there are 12 contiguous cases; and in Tacámbaro (082), the 13 cases are distributed in only 4 nearby zip codes. Furthermore, in Hidalgo (34), an area with a low density of avocado orchards, 13 of the 14 cases are in contiguous zip codes. Outside the avocado agricultural frontier, in Huetamo (038), 11 of the 13 cases are in the same zip code; in Múgica (055), 7 of the 9 cases are in the same zip code; in La Huacana (035), 4 of

**Table 8  Municipalities coded according to the state map, with expected and observed cases of leukemia.**

| No. | Municipality | Expected cases | Observed cases | No. | Municipality | Expected cases | Observed cases |
|---|---|---|---|---|---|---|---|
| 001 | Acuitzio | 0.93 | <5 | 058 | Nuevo Parangaricutiro | 1.66 | <5 |
| 002 | Aguililla | 1.33 | <5 | 059 | Nuevo Urecho | 0.65 | <5 |
| 003 | Álvaro Obregón | 1.74 | <5 | 060 | Numarán | 0.70 | <5 |
| 004 | Angamacutiro | 1.10 | 0 | 061 | Ocampo | 2.31 | 0 |
| 005 | Angangueo | 0.91 | <5 | 062 | Pajacuarán | 1.62 | <5 |
| 006 | Apatzingán | 10.58 | 11 | 063 | Panindícuaro | 1.19 | 0 |
| 007 | Aporo | 0.28 | 0 | 064 | Parácuaro | 2.29 | <5 |
| 008 | Aquila | 2.38 | <5 | 065 | Paracho | 2.92 | <5 |
| 009 | Ario | 3.03 | <5 | 066 | Pátzcuaro | 7.38 | <5 |
| 010 | Arteaga | 1.86 | <5 | 067 | Penjamillo | 1.15 | 5 |
| 011 | Briseñas | 0.88 | 0 | 068 | Peribán | 2.25 | <5 |
| 012 | Buenavista | 3.80 | 5 | 069 | La Piedad | 7.62 | <5 |
| 013 | Carácuaro | 0.83 | <5 | 070 | Purépero | 1.06 | <5 |
| 014 | Coahuayana | 1.16 | 0 | 071 | Puruándiro | 5.49 | <5 |
| 015 | Coalcomán | 1.49 | <5 | 072 | Queréndaro | 1.10 | 5 |
| 016 | Coeneo | 1.39 | 5 | 073 | Quiroga | 2.00 | <5 |
| 017 | Contepec | 2.97 | <5 | 074 | Cojumatlán | 0.78 | <5 |
| 018 | Copándaro | 0.73 | <5 | 075 | Los Reyes | 5.31 | 0 |
| 019 | Cotija | 1.51 | <5 | 076 | Sahuayo | 5.91 | 6 |
| 020 | Cuitzeo | 2.33 | <5 | 077 | San Lucas | 1.31 | 0 |
| 021 | Charapan | 0.96 | 0 | 078 | Santa Ana Maya | 0.93 | 0 |
| 022 | Charo | 1.66 | <5 | 079 | Salvador Escalante | 4.08 | 0 |
| 023 | Chavinda | 0.75 | <5 | 080 | Senguio | 1.67 | 7 |
| 024 | Cherán | 1.56 | <5 | 081 | Susupuato | 0.79 | <5 |
| 025 | Chilchota | 3.08 | <5 | 082 | Tacámbaro | 6.33 | <5 |
| 026 | Chinicuila | 0.41 | <5 | 083 | Tancítaro | 2.82 | 13 |
| 027 | Chucándiro | 0.32 | <5 | 084 | Tangamandapio | 2.51 | <5 |
| 028 | Churintzio | 0.31 | 0 | 085 | Tangancícuaro | 2.51 | <5 |
| 029 | Churumuco | 1.33 | <5 | 086 | Tanhuato | 1.17 | <5 |
| 030 | Ecuandureo | 0.84 | <5 | 087 | Taretan | 1.11 | 0 |
| 031 | Epitacio Huerta | 1.39 | 0 | 088 | Tarímbaro | 7.87 | <5 |
| 032 | Erongarícuaro | 1.13 | <5 | 089 | Tepalcatepec | 1.81 | <5 |
| 033 | Gabriel Zamora | 1.82 | <5 | 090 | Tingambato | 1.18 | <5 |
| 034 | Hidalgo | 10.21 | 14 | 091 | Tinguindín | 1.06 | <5 |
| 035 | La Huacana | 2.82 | 6 | 092 | Tiquicheo | 1.26 | <5 |
| 036 | Huandacareo | 0.81 | 0 | 093 | Tlalpujahua | 2.46 | <5 |
| 037 | Huaniqueo | 0.49 | 0 | 094 | Tlazazalca | 0.41 | <5 |
| 038 | Huetamo | 3.09 | 13 | 095 | Tocumbo | 0.83 | 0 |
| 039 | Huiramba | 0.71 | 0 | 096 | Tumbiscatío | 0.66 | <5 |
| 040 | Indaparapeo | 1.38 | <5 | 097 | Turicato | 2.78 | 0 |
| 041 | Irimbo | 1.26 | <5 | 098 | Tuxpan | 2.23 | <5 |

**Table 8** (*continued*)

| No. | Municipality | Expected cases | Observed cases | No. | Municipality | Expected cases | Observed cases |
|-----|--------------|----------------|----------------|-----|--------------|----------------|----------------|
| 042 | Ixtlán | 0.94 | <5 | 099 | Tuzantla | 1.31 | <5 |
| 043 | Jacona | 5.62 | <5 | 100 | Tzintzuntzan | 1.08 | <5 |
| 044 | Jiménez | 0.87 | <5 | 101 | Tzitzio | 0.82 | 5 |
| 045 | Jiquilpan | 2.33 | <5 | 102 | Uruapan | 25.31 | <5 |
| 046 | Juárez | 1.20 | 0 | 103 | Venustiano Carranza | 1.82 | 19 |
| 047 | Jungapeo | 1.84 | <5 | 104 | Villamar | 1.23 | 0 |
| 048 | Lagunillas | 0.44 | <5 | 105 | Vista Hermosa | 1.51 | <5 |
| 049 | Madero | 1.65 | <5 | 106 | Yurécuaro | 2.52 | 0 |
| 050 | Maravatío | 7.36 | <5 | 107 | Zacapu | 5.56 | <5 |
| 051 | Marcos Castellanos | 1.00 | 0 | 108 | Zamora | 15.02 | 5 |
| 052 | Lázaro Cárdenas | 14.21 | 8 | 109 | Zináparo | 0.19 | 9 |
| 053 | Morelia | 51.62 | 45 | 110 | Zinapécuaro | 3.66 | <5 |
| 054 | Morelos | 0.58 | <5 | 111 | Ziracuaretiro | 1.40 | 9 |
| 055 | Múgica | 3.78 | 9 | 112 | Zitácuaro | 13.45 | <5 |
| 056 | Nahuatzen | 2.36 | <5 | 113 | José Sixto Verduzco | 1.85 | 6 |
| 057 | Nocupétaro | 0.72 | <5 | | | | 0 |

the 6 cases are in the same zip code. In Apatzingán (006), 8 of 10 cases are in a contiguous area despite being spread across 7 zip codes. Likewise, for Zamora (108), which has 9 cases, and Jacona (43), which has 5 cases, the zip codes of the cases were nearby.

## DISCUSSION

Worldwide, one of the main causes of childhood morbidity and mortality is cancer (*Organización Panamericana de la Salud (OPS), 2024*). In México, 2,000 new cases of leukemia are reported each year in children under 18 years of age, making it the leading cause of cancer death in children (*INSP, 2021*). Leukemia represents a serious public health problem, so having epidemiological records of the disease will help to improve programs for timely detection, care, and treatment, based on the characteristics of the Mexican population.

The sociodemographic characteristic predominant in the study population was an age range of 1 to 9 years, with a median age of 6 years and a peak age between 2 and 4 years. These data are consistent with reports in the country (2 to 5 years) (*Shalkow-Facs, 2017*; *García-Mélendez et al., 2023*). As in international reports (*Kakaje et al., 2020*; *Tseng, Lee & Lin, 2023*), the predominant leukemia by lineage was lymphoid (88.3%), specifically B-cell acute lymphoblastic leukemia, while AML was the second most common.

Avocado production in Michoacán has grown rapidly, reaching 186,713 hectares planted by 2023 (*Secretariat of Agriculture, Livestock, Rural Development, Fisheries, and Food (SAGARRA), 2024*). Despite the economic importance, there are few records on the agrochemical use in avocado production or their impact on the health of workers and their families. This underscores the importance of future research on the health impacts experienced by workers exposed to pesticides used in avocado cultivation and their potential implications. Regarding the impact on health, several international reports associate the

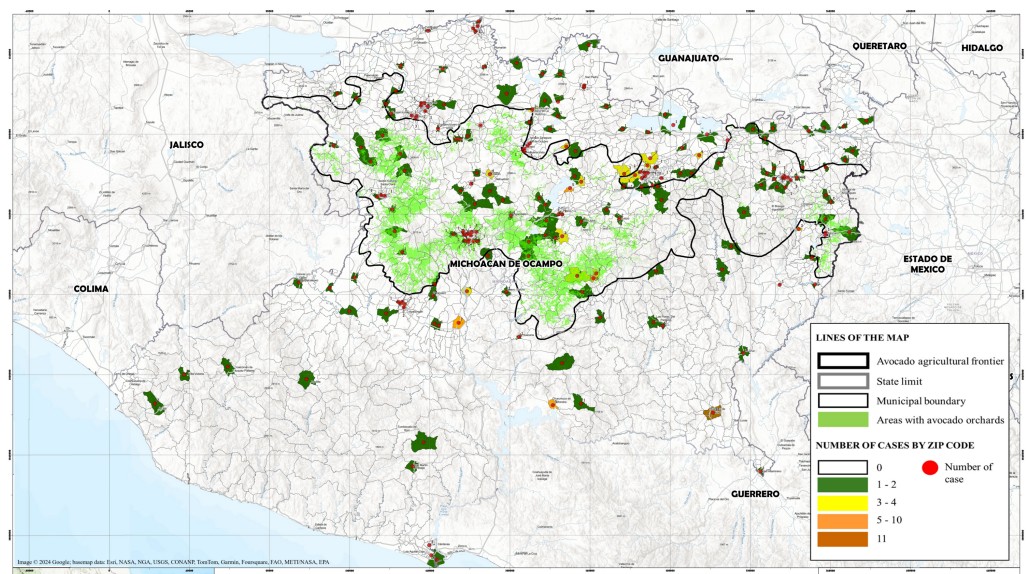

**Figure 5** **Distribution of childhood leukemia cases by municipalities in Michoacán.** Sociodemographic data were obtained from medical records of pediatric leukemia patients treated at the "Eva Sámano de López Mateos" Children's Hospital between 2010 and 2023. Metadata prepared by the University Drone Laboratory (LUD), Center for Research in Environmental Geography (CIGA). Map adapted by the author. All rights reserved by the respective data providers. Map generated using Google Earth Pro (Image ©2024 Google; basemap data: Esri, NASA, NGA, USGS, CONANP, TomTom, Garmin, Foursquare, FAO, METI/NASA, EPA).

use of agrochemicals with childhood leukemias (*Rull et al., 2009*; *Gunier et al., 2017*; *Patel et al., 2020*; *Karalexi et al., 2021*), which makes it essential to identify a possible association between childhood leukemias and avocado farming.

Occupational analysis revealed that only 3.4% of mothers had a direct link to agrochemical use through their profession (Table 3). Among the activities recorded in the medical files were: peasant, cutter or collector, day laborer, and fertilizer sales. Moreover, no maternal occupation showed statistically significant associations with the area of residence (Table 5). Previous studies have documented that maternal exposure to agrochemicals during pregnancy is a risk factor for childhood leukemia (*Karalexi et al., 2021*; *Kumar, Vashist & Rathee, 2014*). However, the medical records reviewed in this study did not include information regarding such exposure. This represents a limitation in understanding the overall risk of leukemia development. On the other hand, paternal occupation involving agrochemical use may pose a risk of secondary exposure within the household. Several studies have indicated that agricultural workers can carry pesticide residues on their clothing, skin, or tools, potentially exposing other family members. Nevertheless, this depends on the type of agrochemical, duration of exposure, and overall hygiene practices of the agricultural worker at home (*López-Gálvez et al., 2019*). This mode of exposure could be particularly relevant during pregnancy. However, in the absence of such data in the medical records, risk assessment based on paternal occupation may offer an indirect approximation of potential exposure within the household or directly to the

child. These findings highlight the need for future research including direct assessments of maternal exposure, supported by detailed surveys on agrochemical use during pregnancy.

In the analysis of paternal occupation, the most frequent work activity was related to agrochemical use, accounting for 46.6% of the total cases. Furthermore, fathers engaged in occupations involving agrochemical use were 1.7 times more likely to reside in the avocado-growing area compared to those with other occupations (OR = 1.764; 95% CI [1.034–3.009]; $p = 0.0379$). This finding suggests a higher concentration of potentially agrochemical-exposed occupations in that area.

Several international studies have linked agrochemical exposure to an increased risk of childhood leukemia. For instance, a study conducted in California identified an association between paternal occupational exposure to pesticides and the risk of childhood acute lymphoblastic leukemia (ALL) (*Gunier et al., 2017*). Similarly, the International Childhood Cancer Cohort Consortium, which compiled data from five countries (Australia, Denmark, Israel, Norway, and the United Kingdom), reported a positive association between paternal exposure to pesticides and a higher risk of childhood acute myeloid leukemia (AML) (*Patel et al., 2020*). Furthermore, residential proximity to areas where agrochemicals are used has been proposed as a potential risk factor for ALL, likely due to moderate lifetime exposure to pesticides (*Rull et al., 2009*). In México, health risks associated with agrochemical exposure have been evaluated in small populations, particularly among agricultural workers, their families, and residents living near crop fields. Studies have documented hormonal and behavioral changes, cellular and hematological alterations (*García-Hernández et al., 2018*; *Silveira-Gramont et al., 2018*), and in some cases, genetic damage linked to cancer. For example, a study analyzing a limited number of samples (nine umbilical cord samples and three cancer cases, including one of leukemia) found mutations in genes such as *P53* and *NOTCH1*, both catalogued in the COSMIC database (*Martínez-Valenzuela et al., 2017*). A 2009 study by the Clinical and Epidemiological Research Unit of IMSS General Regional Hospital No. 20 investigated risk factors associated with acute leukemia in a sample of 47 children. It concluded that paternal pesticide use before pregnancy may represent a potential risk factor for childhood leukemia (*Hernández-Morales, Zonana-Nacach & Zaragoza-Sandoval, 2009*). However, to date, no studies in México have assessed the postnatal risk of childhood leukemia in relation to paternal occupation or residential proximity to agricultural areas where agrochemicals are used. The present study contributes to this knowledge gap by providing preliminary evidence that may support further investigation into these potential environmental and occupational risk factors.

Interestingly, higher survival rates are observed in leukemia patients whose parents are involved in agrochemical-related activities (Fig. 2). Several factors could influence this association, including differences in treatment follow-up, patient-specific sociodemographic characteristics, genetic variants affecting treatment response, or even pre-disease nutritional status. Regarding this last factor, a possible hypothesis for future studies is the impact of the nutritional status of these patients, since it has been reported that high fruit and vegetable consumption can increase survival in patients with head, neck, and ovarian cancer (*Hurtado-Barroso et al., 2020*). Specifically, avocados contain various bioactive compounds with potential anticancer effects, including antimicrobial peptides,

carotenoids, terpenoids, D-mannoheptulose, persenone A and B, phenols, and glutathione (*Dreher & Davenport, 2013*; *Ochoa-Zarzosa et al., 2021*). Furthermore, a risk analysis found that consuming more than one serving of avocado per week is associated with a lower risk of cancer (*Ericsson et al., 2023*). However, the medical records used in the present study lack the detailed nutritional information needed to establish a direct relationship between avocado consumption and childhood leukemia survival. Further research that accounts for multiple influencing factors is necessary to clarify these findings.

The second most common paternal occupation among the leukemia patients under study was related to construction (12.5%), including activities such as bricklayer, construction worker, and construction supervisor. A literature search was conducted to analyze the direct association between parental occupation in construction and the incidence of childhood leukemias. However, the only record of direct association occurs when construction is performed at home, either by remodeling or construction, with a significant risk for children to develop ALL (*Whitehead et al., 2017*), implying that children have direct contact with the construction site. On the other hand, studies on construction workers themselves indicate that there is a risk for workers to develop leukemia (ALL, AML, and chronic myeloid leukemia or CML) (*Luckhaupt et al., 2012*). Additionally, a risk of leukemia has been identified in workers exposed to brick mortar (a mixture of cement, lime, or both used as a binder, sand, and water for building walls) and in those who come into contact with paints and solvents in construction, such as carbamates, diethylene glycol, ethanol, and ethylbenzene in flooring, insulation, and roofing products (*Huang et al., 2022*). In this sense, the use of paints and solvents has already been described as a risk factor for leukemia. It would be necessary to investigate whether construction workers are in contact with these products and if this can represent a risk to their families, including their children, either through direct or indirect contact. Furthermore, there are other chemicals found in construction materials that should be identified to see if they are present in the materials used in the country, which could pose a risk factor for leukemia in construction workers or their children, such as formaldehyde, a carcinogen used as a preservative in wood products or drywall; carcinogenic residual monomers found in polymeric materials like melamine, polyvinyl chloride, ethyleneimine, etc.; and plasticizers like BBP phthalate found in vinyl flooring, among others (*Huang et al., 2022*). Regarding survival, children whose parents were engaged in construction had the lowest cumulative survival rate. It would be interesting to study whether the feeding hypothesis proposed for children whose parents are involved in agriculture could be reversed in this group, assuming that the quality of their food is poorer compared to the agricultural occupation group. This is relevant since it has been identified that there is a direct association between nutritional status at the time of diagnosis of leukemias, lymphomas, and other solid tumors and overall survival. It has been shown that both undernourished and overweight and obese children have worse overall survival (*Karalexi et al., 2022*).

On the other hand, the observed frequency of mortality during treatment (remission induction chemotherapy) was 5.3% ($n = 25$). At the national level, this report is comparable with other studies (*Aguilar-Hernández et al., 2017*), even in reports of early death (death during the first year) (*Martín-Trejo et al., 2017*). In this study, the percentage of early death

(death during the first year after diagnosis) was 10.2% ($n = 44$). However, international reports indicate less than 5% of deaths in the remission induction stage (*Rubnitz et al., 2004*). As in this investigation, septic shock and hemorrhage are reported to be the main causes of death in these patients at the national level [34] and in cancer patients worldwide, accounting for up to 50% of all cancer deaths (*Agulnik, 2023*).

Regarding the number of cases per municipality, five of the six states with an incidence equal to or greater than ten cases belong to the avocado-growing zone (Morelia, Uruapan, Hidalgo, Tacámbaro, and Apatzingán). Uruapan and Tacámbaro, densely occupied by avocado orchards, account for 32 cases (7.9%) in contiguous postal zones. Between Pátzcuaro and Salvador Escalante, another 12 cases are in contiguous zip codes in areas densely occupied by avocado orchards. In Morelia, the state capital, 45 cases were found. Although it is not one of the areas with the highest density of orchards, many cases could be explained by the fact that they attended Children's Hospital. In Hidalgo, with 11 cases, located in an area where, although orchards are visible, it is not one of the most representative, it was found that in 36% of the cases, the parents are engaged in agriculture and another 36% in activities related to carpentry, where contact with paints and solvents could be a determinant explaining the number of cases in this area. Outside the avocado-growing zone, Huetamo has 13 cases, with 61% of the parents dedicated to agriculture, suggesting that the source of agrochemicals for these workers and their families could be maintained. However, the residential proximity (11 of the 13 cases in a single zip code) raises concerns and indicates a need to investigate the origin of these cases. The cases in municipalities such as Múgica and La Huacana, were also concentrated in the same zip code. In Apatzingán, Zamora, and Jacona, although the cases are not in the same zip codes, their proximity underscores the need for a study by municipalities of residence.

This study may have global relevance due to the widespread use of agrochemicals worldwide. The findings on the association between childhood leukemia, paternal occupation, and residential area could be extrapolated to other populations and agricultural contexts. Moreover, by highlighting the associated risks, the research can increase global awareness of potential health hazards.

## STUDY LIMITATIONS

The research presents several limitations that must be considered when interpreting the results. The "Eva Sámano de López Mateos" Children's Hospital concentrates approximately 90% of leukemia cases in the state; however, there is no registry of cases treated at IMSS, ISSSTE, or neighboring states, which could influence the scope of the analyzed data.

Additionally, the records do not accurately reflect the type, duration, or intensity of agrochemical exposure, which could result in misclassification bias. Missing data in medical records may limit the identification of more precise epidemiological patterns. For instance, some unmeasured confounding factors, such as socioeconomic status, nutrition, and follow-up medical care, could influence susceptibility to hematologic diseases.

The absence of this information in medical records highlights the need for future studies focused on evaluating the combined impact of agrochemicals—including type, duration,

and intensity of exposure—and a more comprehensive assessment of health conditions in leukemia patients. Furthermore, measuring biomarkers could help establish a direct relationship between agrochemical exposure and health effects in exposed individuals.

## CONCLUSIONS

The research presented here has revealed evidence of the association between childhood leukemia in the state of Michoacán and occupational exposure to agrochemicals in avocado orchards. These findings highlight the need to deepen information on the use of agrochemicals by agricultural workers, as well as to monitor indirect exposure in the families of workers (due to indirect contact or residential proximity). Furthermore, the incidence of leukemia in children of construction workers found in this study underscores the need to conduct studies focused on these workers and their families. Ultimately, this research aims to raise awareness among the population about the health risks posed by the use of agrochemicals in the state, as well as to increase safety and protection measures to reduce harmful health effects.

## ACKNOWLEDGEMENTS

We would like to thank all the oncology staff at the Hospital Infantil de Morelia "Eva Sámano de López Mateos" for their cooperation in searching for and analyzing medical records. We also thank the Health Intelligence Center, which belongs to the Ministry of Health of the state of Michoacán de Ocampo, México, for their kind contribution to complement this work by sharing the mortality records of pediatric leukemia patients corresponding to the study period.

### Funding

This research was funded by the Consejo Nacional de Humanidades, Ciencias y Tecnología (CONAHCyT), México—currently referred to as the Secretaría de Ciencia, Humanidades, Tecnología e Innovación (SECIHTI)—project 322772, and by the Institute of Science, Technology and Innovation (ICTI) of the state of Michoacán de Ocampo, México. This work was funded through support for a postdoctoral researcher with the Unique Curriculum Vitae (CVU) number: 444595. The funders had no role in study design, data collection and analysis, decision to publish, or preparation of the manuscript.

### Grant Disclosures

The following grant information was disclosed by the authors:
Consejo Nacional de Humanidades, Ciencias y Tecnología (CONAHCyT), México.
Secretaría de Ciencia, Humanidades, Tecnología e Innovación (SECIHTI): 322772.
Institute of Science, Technology and Innovation (ICTI) of the state of Michoacán de Ocampo, México.
Unique Curriculum Vitae (CVU): 444595.

## Competing Interests

The authors declare there are no competing interests.

## Author Contributions

- Paola Jiménez-Alcántar performed the experiments, analyzed the data, prepared figures and/or tables, authored or reviewed drafts of the article, and approved the final draft.
- Anel Gómez-García conceived and designed the experiments, analyzed the data, prepared figures and/or tables, authored or reviewed drafts of the article, and approved the final draft.
- Joel E. López-Meza conceived and designed the experiments, performed the experiments, authored or reviewed drafts of the article, and approved the final draft.
- Alejandra Ochoa-Zarzosa conceived and designed the experiments, performed the experiments, authored or reviewed drafts of the article, and approved the final draft.
- Luis Andrés Espino-Barajas analyzed the data, authored or reviewed drafts of the article, and approved the final draft.
- Luis Miguel Morales-Manilla analyzed the data, authored or reviewed drafts of the article, and approved the final draft.
- Eloy Pérez-Rivera performed the experiments, authored or reviewed drafts of the article, and approved the final draft.
- Luz Yadira Zúñiga-Quijano performed the experiments, authored or reviewed drafts of the article, and approved the final draft.
- Sergio Gutiérrez-Castellanos conceived and designed the experiments, performed the experiments, analyzed the data, prepared figures and/or tables, authored or reviewed drafts of the article, and approved the final draft.

## Human Ethics

The following information was supplied relating to ethical approvals (i.e., approving body and any reference numbers):

The Research Committee of the Nuevo Hospital de Morelia 'Eva Sámano de López Mateos' granted ethical approval to carry out the study with the number CI/10/2023. This project was also approved by the Local Health Research Committee 1602 of the Regional General Hospital Number 1 of the Mexican Social Security Institute with the registration number: R-2023-1602-060.

## Data Availability

The data is available in the Supplemental Files.

## Supplemental Information

Supplemental information for this article can be found online at http://dx.doi.org/10.7717/peerj.20219#supplemental-information.

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
