# Peer review of "Retrospective study on the association between paternal occupational exposure to agrochemicals and childhood leukemia in Michoacán de Ocampo, México"

_PeerJ, doi:10.7717/peerj.20219_

## Round 0.1 · original submission · Major Revisions

· Academic Editor

Major Revisions

Reviewer 1 ·

Basic reporting

1. The manuscript lacks clarity on the specific chemicals used in avocado farming. While the agricultural context is explained, no discussion or table summarizes known toxic agents typically applied in avocado orchards based on the literature. This will provide potential mechanistic interpretation and future research directions.

2. Demographic comparisons between cases and controls are mentioned but not clearly presented in tabular form with corresponding Chi-square test results and p-values.

3. The data and statistical outputs supporting logistic regression are insufficiently reported.

Experimental design

1. The study lacks a control group of children without leukemia for clear exposure comparisons.

2. Paternal and maternal occupational exposure are collected, but maternal exposure is only briefly discussed despite known risks during gestation.

3. Children's own exposure (e.g., through residence, take-home exposure, water, or soil) is not directly assessed or discussed in sufficient detail.

4. The exposure variable is too broadly categorized ("agrochemical-related occupation") without distinguishing types of exposure, intensity, or duration.

This information may not be available and should be included as limitations in the discussion.

Validity of the findings

1. A table comparing demographic characteristics between cases and controls (e.g., sex, age, maternal occupation) and p-values from Chi-square tests is missing and should be provided.

2. The odds ratios are mentioned, but a full model output table with variables, reference categories, ORs, CIs, and p-values is needed. It is unclear which group served as the reference category. This must be explicitly stated.

3. The model covariates (adjusted for age only?) seem limited; consideration of sex and socioeconomic status would improve robustness.

4. As you noted, the paper should present ORs for the following four groups:
Children in avocado-growing areas with agrochemical-occupational parents.
Children in avocado-growing areas with non-agrochemical parents.
Children in non-avocado areas with agrochemical parents.
Children in non-avocado areas with non-agrochemical parents (reference group).
This would strengthen the interpretation of interaction effects between residential exposure and occupational exposure.

5. The conclusion that agrochemical exposure is associated with better survival is speculative and should be interpreted cautiously. Alternative explanations (e.g., treatment differences, access to care, data artifacts) should be explored.

Additional comments

1. Include a section or table summarizing agrochemicals typically used in avocado farming with known toxicological profiles linked to leukemia or hematological effects. This would greatly improve biological plausibility.

2. More detailed analysis and discussion of maternal occupation and possible in utero exposure pathways are needed. Consider discussing or suggesting future measurement of take-home pesticide exposure (e.g., through dust, clothes, shoes) and children’s contact with agricultural environments.

3. Acknowledge limitations such as misclassification bias (occupational category from records), missing data, and potential unmeasured confounding (e.g., SES, nutrition, healthcare access).

Reviewer 2 ·

Basic reporting

The manuscript is clearly structured and follows standard sections. Good general writing in English

Experimental design

Suggestions
Exposure to agrochemicals is not quantified (type, duration, intensity). It is recommended to include this point as an important limitation, with proposals for future studies (e.g., biomarkers, detailed surveys).

It would be valuable to describe whether controls were applied for population density or access to medical care in the different municipalities.

Validity of the findings

The finding of greater survival in children of agricultural workers could be confounded by access to fresh food or uncontrolled factors.

Add a section on limitations of the study (for example, selection bias due to the reference hospital).

Additional comments

In the discussion, you can compare more explicitly with similar studies in other agricultural regions of the world.

Mention if there were differences by sex in incidence or survival.

Reviewer 3 ·

Basic reporting

This goal is this study is to examine how paternal occupational exposure to agrochemicals might be associated with childhood leukemia in Michoacan de Ocampo. Overall, the writing is comprehendible but could be strengthened with an English editor. References and background/context were sufficient. The survival results were interesting. Most of my concerns were related to the analyses for risk of leukemia.

Figures and tables look fine; however, text for figure captions are needed. It is not clear what Figures 3 and 4 are displaying. It is hard to read the map legends. Maybe you can display based on incidence rate instead of crude numbers because the crude numbers can be misleading without a comparison group (i.e., number of children in the municipality or zip code at risk during the time period).

Experimental design

The research question was well described but did not include survival outcomes in the Introduction. Please find specific comments that could strengthen the manuscript -

1. Can the authors add some text about the proportion of children with leukemia they think the hospital sees from the region? Do they think they are missing potential leukemia cases from the region?
2. Line 120-122: The sentence about medical records not specifying the duration and type of exposure to agrochemicals seems out of place here.
3. Line 169: Can authors elaborate how they defined “area of residence”?
4. Could the authors estimate the incidence rate of leukemia by their catchment area and by avocado zone? While the high frequency of leukemia cases in the avocado zone is interesting, what if that is also a reflection of a higher proportion of children living in that zone? Alternatively, you could calculate the expected number of cases and compare with the number of you observed.
5. Lines 219-226: I do not fully understand what is being presented. How does the model explain 48% of variation in survival related to the first sentence? Perhaps it is misplaced? Are the results correlating paternal occupation with avocado growing zone? A father in an agrochemical occupation is more likely to live in the avocado growing zone than construction or paints and solvents? I don't fully understand what the comparison group is. Maybe create a table with the Ns, ORs, and p-values?
6. Survival analyses - did the authors consider a Cox regression model so they can adjust for potential confounders (eg., age at diagnosis, sex, etc). Differences in survival by sex have been reported in other countries. Did the authors see that here?

Validity of the findings

I have no concerns with describing the study population. Please refer to previous comments about some concerns related to the Methods and Results presented.

---

## Round 0.2 · Minor Revisions

· Academic Editor

Minor Revisions

Reviewer 1 ·

Basic reporting

I just have a few minor editing comments:

Lines 208–210: Please revise this section as complete sentences and a paragraph, not bullet points.

Tables: Remove bullet points from Table 1.

All tables are currently formatted differently; please make the formatting consistent across all tables.

For Table 2, add a header for the first two columns labeled "Characteristics."

Some pages have larger fonts while others have smaller fonts; please standardize the font size and formatting throughout the document.

Experimental design

-

Validity of the findings

-

Additional comments

The authors have responded to the reviewer comments thoroughly and revised the manuscript appropriately.

Reviewer 3 ·

Basic reporting

Thank you for the thoughtful response to the reviewers' comments, which have improved the quality of the manuscript. The manuscript is overall clear, cites sufficient references, and has self-contained hypotheses. I just have a couple of minor suggestions for your consideration. Details provided below.

Experimental design

-Please consider noting the tables presenting OR or HRs if the results are from unadjusted or adjusted models. If adjusted, please list the variables in a footnote.

-Sorry if I missed this, but how were the expected cases calculated in Table 4?

-Table 8 - Are the authors concerned about potential loss of patient confidentiality by reporting the exact number of observed cases in each municipality when there are fewer than 5 cases? This will be IRB-dependent. From my experience, if there are concerns about reporting the exact number of cases, we typically report "<5".

Validity of the findings

-

---

## Round 0.3 · Minor Revisions

· Academic Editor

Minor Revisions

This is a minor issue and a bit required after additional review.

Reviewer 1 ·

Basic reporting

The authors have responded to the reviewers’ comments and revised the manuscript clearly. I am satisfied with the revisions and have no further concerns.

Experimental design

The authors have responded to the reviewers’ comments and revised the manuscript clearly. I am satisfied with the revisions and have no further concerns.

Validity of the findings

The authors have responded to the reviewers’ comments and revised the manuscript clearly. I am satisfied with the revisions and have no further concerns.

Additional comments

Lines 384–392 should be presented as a paragraph using complete sentences rather than as a numbered list.

Reviewer 3 ·

Basic reporting

-

Experimental design

When calculating the expected number of cases for the SIR, the authors stated that they "multiplied the corresponding average population by the state leukemia incidence rate." Can the authors specify whether an age cutoff was used for the average population? I assume the authors used the average population of 0–17-year-olds. Is there a year this number came from, or what data source provided the average population estimate?

Validity of the findings

-

---

## Round 0.4 · accepted · Accept

· Academic Editor

Accept

This revised version is suitable for publication in PeerJ.